# Carbon-Enhanced Hydrated Salt Phase Change Materials for Thermal Management Applications

**DOI:** 10.3390/nano14131077

**Published:** 2024-06-24

**Authors:** Yizhe Liu, Xiaoxiang Li, Yangzhe Xu, Yixuan Xie, Ting Hu, Peng Tao

**Affiliations:** 1State Key Laboratory of Metal Matrix Composites, School of Materials Science and Engineering, Shanghai Jiao Tong University, 800 Dong Chuan Road, Shanghai 200240, China; 780145898@sjtu.edu.cn (Y.L.); lixiaoxiang@sjtu.edu.cn (X.L.); xuyz1998@sjtu.edu.cn (Y.X.); xyx0408@sjtu.edu.cn (Y.X.); ht-huting@sjtu.edu.cn (T.H.); 2Materials Genome Initiative Center, School of Materials Science and Engineering, Shanghai Jiao Tong University, 800 Dong Chuan Road, Shanghai 200240, China; 3National Engineering Research Center of Special Equipment and Power System for Ship and Marine Engineering, Shanghai 200030, China

**Keywords:** hydrated salt, phase change material, carbon, thermal storage, thermal management

## Abstract

Inorganic hydrated salt phase change materials (PCMs) hold promise for improving the energy conversion efficiency of thermal systems and facilitating the exploration of renewable thermal energy. Hydrated salts, however, often suffer from low thermal conductivity, supercooling, phase separation, leakage and poor solar absorptance. In recent years, compounding hydrated salts with functional carbon materials has emerged as a promising way to overcome these shortcomings and meet the application demands. This work reviews the recent progress in preparing carbon-enhanced hydrated salt phase change composites for thermal management applications. The intrinsic properties of hydrated salts and their shortcomings are firstly introduced. Then, the advantages of various carbon materials and general approaches for preparing carbon-enhanced hydrated salt PCM composites are briefly described. By introducing representative PCM composites loaded with carbon nanotubes, carbon fibers, graphene oxide, graphene, expanded graphite, biochar, activated carbon and multifunctional carbon, the ways that one-dimensional, two-dimensional, three-dimensional and hybrid carbon materials enhance the comprehensive thermophysical properties of hydrated salts and affect their phase change behavior is systematically discussed. Through analyzing the enhancement effects of different carbon fillers, the rationale for achieving the optimal performance of the PCM composites, including both thermal conductivity and phase change stability, is summarized. Regarding the applications of carbon-enhanced hydrate salt composites, their use for the thermal management of electronic devices, buildings and the human body is highlighted. Finally, research challenges for further improving the overall thermophysical properties of carbon-enhanced hydrated salt PCMs and pushing towards practical applications and potential research directions are discussed. It is expected that this timely review could provide valuable guidelines for the further development of carbon-enhanced hydrated salt composites and stimulate concerted research efforts from diverse communities to promote the widespread applications of high-performance PCM composites.

## 1. Introduction

Thermal energy, which accounts for ~50% of energy end-use, is the cornerstone of the global energy supply and consumption chain [1]. For example, keeping houses within a comfortable temperature range regardless of climate change and generating hot water consume about half of the overall energy used by buildings [2]. To date, more than 70% of consumed thermal energy comes from the burning of fossil fuels, which calls for the urgent exploration of cleaner renewable energy alternatives [3,4]. The energy supply from renewable sources such as the Sun, however, is not stable and the energy power density is often quite low [5,6]. Addressing the intermittency issue is a prerequisite to developing advanced renewable heating technologies. Another fact is that a large portion of thermal energy is wasted during various energy conversion processes [7], which not only sacrifices energy utilization efficiency but also poses safety concerns. Undesired temperature changes, either rising or dropping, adversely affect the performance of many electronic devices [8,9,10]. How to manage excessive heating and stabilize these devices within the desired temperature range without consuming extra energy has attracted increasing attention.

Solid–liquid phase change materials (PCMs), which can absorb a large amount of heat during their melting processes and release the heat in the subsequent solidification processes [10,11,12], offer a promising solution to overcome the intermittency of renewable energy and smartly manage the excessive heating of electronic devices [13]. In recent years, inorganic hydrated salts have garnered tremendous attention owing to their large specific heat storage capacity, suitable and tailorable phase change temperatures, and notable advantages in cost-effectiveness and nonflammability over their organic counterparts [14,15,16,17]. These attractive features spurred the exploration of hydrated salts for the phase change management of thermal energy, including both harvesting and storage of renewable thermal energy and controlling the temperature of buildings, electronic devices and human bodies [2,18,19,20,21]. Hydrated salts, however, suffer from several common shortcomings including low thermal conductivity, supercooling, poor heating/cooling cycling stability, low solar absorptance, a tendency to phase separate, leakage and loss of crystallization water and strong corrosiveness [16,22,23,24,25]. Compounding hydrated salts with functional fillers has been pursued as the mainstream route to address these issues [26,27].

Among the various fillers investigated so far, carbon materials have emerged as one of the favorites for comprehensively improving the thermophysical properties of hydrated salts because they possess various desired features, such as high thermal conductivity, strong solar absorptance, large specific surface area, light weight, and an abundance of low-cost raw materials [16,23,28,29,30]. In recent years, attempts have been made to compound hydrated salts with different carbon fillers, including one-dimensional (1D) carbon nanotubes (CNTs) and carbon fibers (CFs); two-dimensional (2D) graphene oxide (GO), reduced graphene oxide (rGO) and graphene; and three-dimensional (3D) expanded graphite (EG), biochar and activated carbon (Figure 1). Compared with pristine hydrated salts, the carbon-enhanced hydrated salt composites have demonstrated improved thermophysical properties, including high thermal conductivity, suppressed supercooling and strong resistance to phase separation and liquid leakage. The carbon-enhanced hydrated salt PCM composites combine the advantageous features from both the low-cost inorganic PCM components and the abundant carbon fillers to fulfill the potential for diverse thermal-related applications. Through introducing other functional fillers, the carbon-enhanced hydrated salt composites have exhibited extra functionalities, such as the direct conversion and storage of electro-thermal and magneto-thermal energy and dynamic tuning of the thermal energy charging and discharging processes. Meanwhile, the thermal conductivity, specific surface area, and surface chemistry vary for different carbon fillers, which in turn affect their compounding with hydrated salts and the enhancement of thermophysical properties. The selection of appropriate types of carbon fillers, loading levels and compounding processes is critical to achieving the optimal performance of carbon-enhanced hydrated salt PCM composites.

Despite the fact that recent years have witnessed the rapid development of carbon-enhanced hydrated salt PCM composites, a systematic review to summarize how different category of carbon fillers (1D, 2D, 3D and hybrid) enhance the thermophysical properties of hydrated salts and enable their promising applications is lacking. To bridge this gap, herein we provide a comprehensive review of the state-of-the-art development of carbon-enhanced hydrated salt PCM composites, aiming at highlighting the design principles, fabrication strategies, governing enhancement mechanisms and application potentials. We firstly introduce the solid–liquid phase change process and the advantageous features and shortcomings of hydrated salts for the storage and management of thermal energy. We then briefly describe how various carbon materials can enhance the thermophysical properties of hydrated salts and the typical methods for fabricating carbon-enhanced hydrated salt PCM composites. By examining representative examples, we discuss how the intrinsic properties of various carbon materials, such as thermal conductivity, surface chemistry, morphology, microstructure and their loading and dispersion affect the performances of the resultant hydrated salt composites, including heat transfer behavior, reversible solid–liquid phase change stability, anti-leakage, solar–thermal conversion, and additionally afforded functionalities. Regarding the applications of carbon-enhanced hydrated salt composites, we focus on the thermal management of electronic devices, buildings and human bodies by taking advantage of their suitable phase change temperature ranges. Finally, we point out current research needs and future research directions to clarify enhancement mechanisms, develop more advanced fabrication techniques, comprehensively improve overall thermophysical properties and explore diverse applications.

## 2. Hydrated Salt PCMs

Hydrated salts, which can be generally expressed as M_p_X_q_·nH_2_O, rely on the reversible melting–solidification phase transition to realize large-capacity storage and the release of thermal energy within narrow temperature ranges. As a type of PCMs for the regulation of thermal energy, hydrated salts should simultaneously possess a suitable phase change temperature, large latent heat storage capacity, high thermal conductivity, stable solid–liquid phase change behavior and cost advantages, as well as good physical, chemical and thermal stability [16,17,31].

As listed in Table 1, common hydrated salt PCMs have melting temperatures in the range of 8–120 °C, which is suitable for the storage and management of thermal energy at low-to-medium temperature ranges. For example, the low-temperature calcium chloride hexahydrate (CaCl_2_·6H_2_O) with a melting point of 11.2 °C has found important applications in cold storage systems [32]. Other hydrated salts with a melting point around room temperature, such as disodium phosphate dodecahydrate (Na_2_HPO_4_·12H_2_O), have often been used for body heating applications [33]. Moreover, different hydrated salts can be compounded together to form eutectic PCMs and their phase change temperatures can be broadly tuned through altering their chemical compositions [34,35,36]. The melting points of eutectic hydrated salts are still typically lower than 100 °C, which is limited by the thermal loss of the crystallization water. Hydrated salt PCMs usually have a phase change enthalpy within the range of 100–300 kJ/kg and have a larger volumetric latent heat storage capacity (45–120 kWh/m^3^) than organic PCMs, such as paraffin wax (45–60 kWh/m^3^), due to their larger density. High thermal conductivity is crucial to facilitate effective heat exchange during charging/discharging processes and maintain uniform temperature distribution throughout the storage process [37]. Compared with organic PCMs, that most frequently have a low thermal conductivity of ~0.2 W/(m·K), the thermal conductivity of hydrated salts is generally higher and some of them can be larger than 1 W/(m·K) [38], but this is still not sufficient to achieve fast charging and discharging. Other notable advantages from hydrated salts over organic PCMs include material abundance, cost-effectiveness and inflammability, which are highly desired traits for potential large-scale industrial deployment.

The reversible solid–liquid phase transition enables the storage and release of thermal energy. For organic PCMs such as paraffin wax, their phase transition involves a reversible change of the internal molecular arrangement from an ordered crystalline state to a disordered amorphous state when it is over the melting/freezing point. Owing to simple intermolecular interactions during the phase transition processes, organic PCMs often demonstrate congruent melting and solidification, a narrow melting/freezing temperature range, and no phase separation over repeated melting/solidification cycles. By comparison, the phase transition process of hydrated salts is much more complex because of the co-existence of electrostatic interaction and hydrogen bonding between salt ions and water molecules and the potential loss/gain of water molecules [54]. Hydrates can undergo partial melting in the charging process, and most frequently show strong supercooling in the discharging process [55,56,57]. In the melting process, ideally they undergo congruent melting, in which the hydrated salts release all the contained crystallization water and the dehydrated salts fully dissolve in the water, forming an aqueous solution. In many cases, semi-congruent or incongruent melting occurs, meaning that the hydrated salts are not fully dehydrated in the single step and the stoichiometric water is not sufficient to dissolve the dehydrated salts [58,59].

For hydrated salts that undergo two-step melting [54], their melting routes can be described as:(1)$$M_{p}X_{q}\cdot nH_{2}O(s)\to M_{p}X_{q}\cdot mH_{2}O(s)+(n-m)H_{2}O(l)$$
(2)$$M_{p}X_{q}\cdot mH_{2}O\left(s\right)+\left(n-m\right)H_{2}O\left(l\right)\to pM^{q+}\left(aq\right)+qX^{p-}\left(aq\right)+nH_{2}O\left(l\right)$$

One example is that dihydrates form during the melting of Mg(NO_3_)_2_·6H_2_O and the undissolved dihydrates or anhydrous salts cause phase separation. The precipitated salts cannot recombine with water in the subsequent freezing process, which prevents the release of latent heat (Figure 2). During the melting process, the crystallization water can be transformed into vapor, especially when the hydrated salts are rapidly and intensively heated up above their melting points. For example, when sodium acetate trihydrate (CH_3_COONa·3H_2_O, SAT), which has a melting point of 58 °C, is heated to 90 °C, it firstly undergoes a solid–liquid phase change and then experiences a more pronounced heat absorption involving the further decomposition of the crystallization water [60,61]. This will cause an obvious loss of water and lead to phase separation, incongruent melting, and deteriorated storage performances. In addition to chemically bound crystallization water, hydrated salts can adsorb water from the ambient environment, which adds more complexity to the phase transition process.

In the freezing process, many hydrated salts suffer from strong supercooling because of the large energy barrier that must be overcome for formation of a stable crystalline structure [62]. For instance, SAT can be cooled down to −40 °C without crystallization occurring. The large supercooling degree means that the hydrated salt solutions needed to be cooled down to temperatures much lower than the melting point before undergoing crystallization, which impedes the releasing of latent heat and lowers the amount of releasable heat [63].

Like other PCMs, salt hydrates also have a leakage issue when they are melted, and the leaked salt solutions are highly corrosive [25]. Meanwhile, hydrated salts suffer from an inability to be used for direct harvesting of solar-thermal or electro-thermal energy because they are not good solar absorbers or good electrical conductors [23].

## 3. Carbon-Enhancement Strategy

To overcome the shortcomings of pristine hydrated salt PCMs, including low thermal conductivity, phase separation, supercooling, leakage and poor solar absorptance, intensive research efforts have been devoted to compounding them with other functional fillers [64]. As shown in Figure 3, carbon materials have emerged as a favorite filler because they simultaneously possess high thermal conductivity, high solar absorptance, low density, large specific surface area, tunable surface chemistry and good thermal stability [65]. For instance, 1D CNTs and CFs exhibit exceptional high thermal conductivity properties, with axial thermal conductivities of 6600 W/(m·K) and 900 W/(m·K), respectively [66]. Two-dimensional graphene and its derivative have shown high thermal conductivities of 3000–5000 W/(m·K). According to the rule of mixture, after compounding these carbon fillers, the effective thermal conductivity of the hydrated salt composites should be significantly increased at relatively low loading levels. In particular, the heat transfer performance of the composites should be obviously enhanced if these added carbon fillers can form connected networks. The large specific surface area of carbon materials, together with their abundant oxygen-containing surface functional groups, can form strong intermolecular interactions (hydrogen bonding, van der Waals attraction) with melted hydrated salts. The capillary adsorption effect and intermolecular attraction forces can not only help confine and adsorb the melted salts to prevent leakage, but also suppress the loss of crystalline water, thus mitigating phase separation during repeated phase change processes. During the solidification process, these carbon fillers can provide numerous heterogeneous nucleation sites to reduce the supercooling of hydrated salts and to facilitate the liquid-to-solid phase transition and the release of latent heat [67,68,69,70,71]. Three-dimensional carbon, such as graphite foam, can retain the high thermal conductivity traits of carbon materials, and take advantage of its porous structure to inhibit phase separation, leakage, and supercooling issues [17,23]. The rough surface structure and light-trapping feature of 3D carbon materials further amply solar absorptance and provide the resultant hydrated salt composites with efficient solar–thermal conversion capability [71,72,73,74]. Moreover, different types of carbon materials and other functional fillers can be hybridized to collaboratively enhance the thermophysical properties of hydrated salts and impart the PCMs with other functionalities such as electrical conductivity and magnetic responses [27,64].

In general, the preparation routes of carbon-enhanced hydrated salt PCM composites can be classified into two categories (Figure 4), although various approaches and techniques had been investigated. The first type is the carbon additive method, which typically involves the direct physical blending of hydrated salts with carbon materials. In the fabrication process, carbon materials are mixed with melted hydrated salts or solid hydrated salt powders through physical blending, melt blending or solution evaporation preparation methods together with other facilitating techniques such as mechanical mixing, magnetic stirring, ultrasonic dispersing or super-probe dispersing. In these cases, various carbon fillers with a relatively low loading are dispersed within hydrated salts. The obtained composites largely retain the original latent heat storage capacity and processability of the hydrated salts and have cost advantages due to the low loading requirement.

The second type, summarized as the carbon skeleton route, involves loading the molten hydrated salts into 3D carbon skeletons through direct impregnation or vacuum impregnation or compounding solid hydrated salt powders with carbon materials through compression molding processes. In these cases, porous graphite foam, activated carbon and biochar can directly adsorb melted hydrated salts, or compression-molded EG sheets with high loadings can form connected networks to encapsulate hydrated salts. In comparison, the interconnected skeletons are more effective in improving the thermophysical properties of hydrated salts than the sparsely distributed carbon fillers, but the carbon loading is higher and the fabrication process is more complicated for the skeleton method.

## 4. Carbon-Enhanced Hydrated Salt PCM Composites

Diverse carbon materials have been investigated to improve the overall performance of hydrated salts, including their thermal conductivity, supercooling behavior, anti-leakage capability, cycling stability and photothermal conversion. Each carbon material has shown distinct enhancement effects and mechanisms due to its unique features and thermophysical properties. Herein, we provide representative examples to introduce how 1D, 2D, 3D and hybrid carbon materials enhance the thermophysical properties and application performances of hydrate salt PCMs.

### 4.1. Hydrated Salt–CNT Composites

CNTs are tubular structures formed by carbon atoms arranged in a hexagonal lattice similar to that of a graphite sheet, possessing high strength, electrical conductivity and thermal conductivity. Based on the arrangement of carbon atoms in the tubular lattice, CNTs can be classified into single-walled carbon nanotubes (SWCNTs), consisting of a single layer of graphene rolled into a seamless cylinder and multi-walled carbon nanotubes (MWCNTs) consisting of multiple concentric cylinders of graphene [75]. The abundant delocalized π electrons on the surface of CNTs can interact with PCMs through intermolecular forces or chemical bonding, thereby modulating the properties of these materials [76]. With the rapid development of synthesis techniques, such as chemical vapor deposition (CVD), arc discharge and laser ablation, the cost of CNTs, especially MWCNTs, has witnessed a significant drop and can be comparable to the price of some carbon blacks [77]. The control of the loading, dispersion state and quality of CNTs in terms of their high aspect ratio, orientation and surface chemistry is critical for optimizing the enhancement of thermophysical properties of hydrated salts.

By employing various commercial carbon materials as the filler, Kumar et al. [78] investigated the influence of the size and shape of carbon materials on the thermal conductivity enhancement of Mg(NO_3_)_2_·6H_2_O (Magnesium nitrate hexahydrate, MNH) through mixing carbon spheres, MWCNTs, mesoporous carbon, graphene nanosheets, and nano-graphite with the hydrated salts with the same loading of 0.5 wt%, each. The thermal conductivity of the CNT composites was increased by 82.5%, which was higher than the composites loaded with mesoporous carbon (75%), graphene nanosheets (72.5%) and nano-graphite (65%) (Figure 5a). Although CNTs have a high intrinsic thermal conductivity and a large aspect ratio, they found that carbon spheres had stronger enhancement of thermal conductivity (100%) due to their better dispersion and formation of percolated networks. In a recent work, Zeng et al. [79] reported that after hydrophilic treatment, MWCNTs could be uniformly dispersed within the eutectic salt of sodium acetate trihydrate/sodium monohydrogen phosphate dodecahydrate with a mass ratio of 9:1 (Figure 5b). The thermal conductivity of the composites linearly increased with the loading of MWCNTs and reached 1.29 W/m K with 8 wt% of MWCNTs.

Jia’s group [80] developed a CVD method and successfully synthesized aligned CNTs (a-CNTs) by using propylene as the carbon source (Figure 5c), which decomposed on vermiculite-supported catalysts in a fluidized bed reactor. Compared with randomly disordered CNTs (r-CNTs), they reported that these a-CNTs have shown stronger enhancement of the thermophysical properties of organic PCMs. In a recent work [81], they directly blended 5 wt% of a-CNTs with a length of 100 μm and 1 wt% of sodium carboxymethyl cellulose (SCMC) thickening agents to improve the thermal storage performance of binary hydrated salts of Al_2_(SO_4_)_3_·18H_2_O/FeSO_4_·7H_2_O with a mass ratio of 2:1 (Figure 5d). They found that the a-CNTs established a network within the composites, which achieved a thermal conductivity of 3.23 W/(m·K) at room temperature. With increasing temperature, the thermal conductivity slightly decreased due to intensified lattice vibration, but it further rose when the temperature reached 200 °C, which has been attributed to enhanced connectivity between the CNTs after the melting of the cellulose binder. Meanwhile, the CNTs provide a resilient network restraining the volume change of the hydrate salt PCMs during the heat release and absorption processes, thereby enhancing the cycling stability and maintaining a high heat-releasing temperature of 118.43 °C and a large energy storage density of 420.4 J/g after 100 cycles.

In an SAT composite, Sun et al. [82] reported that the addition of 4 wt% of urea could inhibit phase segregation, and 4 wt% of disodium hydrogen phosphate dodecahydrate (Na_2_HPO_4_·12H_2_O, DHPD) nucleation agents could reduce the supercooling degree to 0.6 °C. To improve thermal conductivity and prevent leakage, they compounded the PCMs with mixed EG and CNTs. Interestingly, they found that the addition of 2 wt% of CNTs could further increase the thermal conductivity of the composites from 6.141 W/(m·K) to 7.178 W/(m·K) because they filled the gaps between the hydrated salts and the pores of the EG.

In addition to the enhancement of thermal conductivity, Ding et al. [83] reported that MWCNTs together with SCMC altered the phase change temperature and the fusion enthalpy of DHPD. In contrast to pure DHPD, which exhibited two melting peaks at 33.42 °C and 45.85 °C, the DHPD–SCMC–MWCNT composites effectively suppressed the phase separation of DHPD and the formation of low-grade salt Na_2_HPO_4_·7H_2_O, showed a single melting peak, and increased the latent heat enthalpy by 36.19%.

### 4.2. Hydrated Salt–CF Composites

CF is a material composed of thin, strong crystalline filaments of carbon atoms bonded together in a microcrystalline structure. CF is known for its exceptional strength-to-weight ratio, low density, corrosion resistance and good thermal conductivity. Previously, CFs have found important applications in the aerospace, automotive and construction industries, mainly for enhancing mechanical properties. In recent years, attempts also have been made to incorporate CFs within hydrated salts to improve thermal storage performance [26].

Aiming to improve the thermal properties of DHPD and eliminate phase separation and supercooling, Tang et al. [84] prepared PCM composites with optimized loadings of functional fillers (4 wt% of Na_2_SiO_3_ 9H_2_O, 4 wt% of CMC, 12 wt% of surface-modified AlN and 6 wt% of short-cut CFs) through simple mechanical mixing and vacuum adsorption. In this case, Na_2_SiO_3_ 9H_2_O and surface-modified AlN synergistically inhibit the supercooling of DHPD. Both the surface-modified AlN and short-cut CFs increased the effective thermal conductivity of DHPD hydrated salts due to their intrinsic high thermal conductivity. It was found that 8 wt% of short-cut CFs further improved the thermal conductivity of the composites (PCM-12) at 35 °C from 1.4 W/(m·K) to 1.91 W/(m·K) (Figure 6a). Another advantage from the added CFs was that they provided sufficient capillary confinement to fully eliminate leakage of melted DHPD. In addition to thermal properties, the influence of added carbon fillers on the electrical properties needs to considered if the composites are targeted for the thermal management of electronic devices and systems such as batteries. Unlike surface-modified AlN, CFs are electrically conductive and can form conductive networks within hydrated salts at relative low loadings due to their large aspect ratio. Under applied voltages varying from 5 V to 30 V, the electrical resistance of the composites decreased with increasing loading of CFs. In this case, an optimum loading of 6 wt% was selected to balance the overall thermophysical properties of the composites.

To take advantage of the notable mechanical enhancement effect from CFs, Wang et al. [85] prepared lightweight carbon aerogel through mixing and directional freeze-drying of GO, SCMC and CFs followed by carbonization at relatively low temperatures. They emphasized that added CFs provided structural support for the aerogel and decreased the volume shrinkage of the skeleton. The obtained hydrophilic aerogel with a porosity of 96.9% was loaded with eutectic salts of Na_2_SO_4_·10H_2_O and Na_2_CO_3_·10H_2_O with a mass ratio of 9:1 through melt mixing and vacuum impregnation (Figure 6b). The aerogel helped increase the thermal conductivity of the eutectic salts from 0.544 W/(m·K) to 1.756 W/(m·K), and reduce the supercooling degree from 1.2 °C to 0.7 °C. The enhanced thermal conductivity and reduced supercooling were attributed to the light-weight hydrophilic carbon hydrogel networks. In particular, the low-temperature carbonization process avoided the full loss of hydrophilic functional groups from GO surfaces, such that the aerogel networks maintained good compatibility with the hydrated salts and served as favorable heterogeneous nucleating centers during the solidification process.

### 4.3. Hydrated Salt–GO Composites

Two-dimensional layered carbon, including graphene and its derivatives such as GO and rGO, are extensively utilized for improving the thermal conductivity, phase transition behavior, stability and solar absorptance of hydrated salt PCMs due to their intrinsic high thermal conductivity, large specific surface area, excellent photothermal conversion capability and rich surface functional groups [75]. Among them, GO can be readily fabricated through exfoliation and oxidization of graphite by Hummer’s method at low cost. Various methods have been developed to transform GO into rGO and graphene, which have higher thermal conductivity and stronger photothermal conversion capability [86].

As shown in Figure 7a, Li et al. [67] reported that the addition of GO could reduce the contact angle between melted DHPD and the porous expanded vermiculite (EV) skeleton from 56° to 45° due to the abundance of hydrophilic carboxyl, hydroxyl and epoxy groups on GO sheets. It was found that these functional groups could form hydrogen bonding with crystallization water in hydrated salts, thus preventing the loss of crystallization water and increasing the amount of encapsulated PCMs inside the EV. As a result, the latent heat increased from 167 J/g for the DHPD/EV composites to 229 J/g for the DHPD–GO/EV composites loaded with 0.2 w% of GO. They also noticed that the hydrated salts within DHPD/EV composites were composed of Na_2_HPO_4_·2H_2_O and Na_2_HPO_4_·7H_2_O because of the loss of crystallization water. By contrast, the main components of the hydrated salts within the DHPD–GO/EV composites were Na_2_HPO_4_·7H_2_O and Na_2_HPO_4_·12H_2_O, owing to the water retained by the GO sheets. The formation of hydrogen bonds between salt hydrates and GO also promoted thermal stability, and it took higher temperatures than for the pristine hydrated salts to induce a loss of crystallization water from the composites.

As shown in Figure 7b, Yang’s team [69] prepared a GO-modified poly(acrylamide-co-acrylic acid) copolymer (PAAAM) hydrogel to improve the thermophysical properties of eutectic hydrated salts (EHS, mixture of Na_2_CO_3_·10H_2_O and Na_2_HPO_4_·12H_2_O with a mass ratio of 2:3). Although the hydrophilic hydrogel network could suppress the leakage of melted hydrated salts, the melting enthalpy dropped from 220.2 J/g for the EHS to 189.1 J/g for the PAAAM–EHS composites. Interestingly, it was found that the melting enthalpy of GO–EHS/PAAAM increased to 200.3 J/g, which was ascribed to the weakened electrostatic interaction between PAAAM and EHS by the GO nanosheets. At the same time, the GO–EHS/PAAAM composites showed the lowest supercooling degrees because the hydrophilic GO nanosheets provided numerous favorable heterogeneous nucleation sites to lower the nucleation energy barrier. As a result, the crystallization enthalpy increased from 147.1 J/g for EHS/PAAAM to 160.6 J/g for GO–EHS/PAAAM composites. They also observed an obvious enhancement of thermal conductivity when the GO content exceeded 0.5 wt%, such that the GO sheets could form heat transfer networks, and they achieved a 54% increase in thermal conductivity with addition of 2 wt% GO.

As shown in Figure 7c, Yu’s group [87] reported the fabrication of super-hydrophilic rGO aerogels modified by konjac glucomannan (KGM) through a hydrothermal reaction-freeze-drying process and impregnation of SAT within the skeleton. Owing to the abundant oxygen-containing functional groups in the KGM and rGO surfaces, the aerogels demonstrated excellent wettability with the melted SAT and could effectively prevent the leakage of melted hydrated salts. One notable observation is that unlike conventional fragile rGO aerogels, the KGM-modified aerogels were elastic and could quickly recover to their original state after releasing the applied compression force. The hydrophilic surface treatment also helped increasing the latent heat storage density from 231.6 J/g for the SAT/rGO composites to 252.8 J/g. The aerogel-encapsulated SAT composites enabled direct solar–thermal conversion and accelerated charging and discharging processes.

Although graphene sheets have a high thermal conductivity of 2600 to 5300 W/(m·K) and full-band absorptance of sunlight, they typically have poor miscibility with hydrated salts due to their lack of hydrophilic surface functional groups [89]. To address this shortcoming, Mehrali and coworkers [88] treated graphene nanoplatelets (GNPs) with 1-pyrenecarboxylic acid in an ethanol solution and blended the obtain hydrophilic GNPs with SAT and sodium phosphate monobasic monohydrate (SPM) nucleating agents (Figure 7d). The carboxylic acid groups on the surfaces of the GNPs could form hydrogen bonds with the water molecules in the hydrated salts and restrict the movement of water. Such hydrogen bonding, together with the large specific surface area of the GNPs, created a barrier against phase separation and liquid leakage. They reported that the hydrophilic GNPs reduced the supercooling degree of SAT and achieved similar values as the SPM nucleating agents. With 5 wt% of GNPs, the thermal conductivity was increased by 114%, and the composites achieved a high photothermal conversion efficiency of 92.6%.

### 4.4. Hydrated Salt–EG Composites

Three-dimensional carbon materials can provide interconnected thermal networks to reduce the thermal resistance between carbon fillers; thus, more effective enhancement of the thermal conductivity of hydrated salts is expected [90]. Accelerated heat transfer throughout the composites enables efficient storage and release of thermal energy in the charging and discharging processes, respectively. On the other hand, the large surface area of the porous skeletons provides more nucleation sites to lower the supercooling degree and effectively confine melted hydrated salts to prepare form-stable composites.

Despite EG having a much higher thermal conductivity than GO sheets, its surface is typically hydrophobic due to the lack of oxygen-containing functional groups, and it has poor wettability with hydrated salts. By modifying EG with an Al_2_O_3_ layer (4.4 wt%) by heterogeneous nucleation and a subsequent heat treatment process, Zhang’s group [91] reported that the water contact could be reduced from 90.7° to 33.9° (Figure 8a). Alternatively, they demonstrated that modifying EG with an organic surfactant (trtonX-100) could also create hydrophilic surfaces, which are critical to achieve a large adsorption capacity of hydrated salts. They found that 1:10 was an optimum mass ratio of TriotonX-100 to EG for achieving the highest adsorption capacity of MgCl_2_·6H_2_O-NH_4_Al(SO_4_)_2_·12H_2_O eutectic salt hydrates (81.9%). Compared with eutectic salt hydrates, the thermal conductivity was increased by 10 times, reaching 4.789 W/(m·K), and the composites demonstrated excellent form-stability and thermal reliability.

As shown in Figure 8b, Li et al. [92] transformed expandable graphite into EG through high-temperature treatment at 800 °C and impregnated the eutectic hydrated salts of sodium carbonate decahydrate (Na_2_CO_3_·10H_2_O, SCD) and SDPD to prepare form-stable composites. The EG network helped decrease the supercooling degree of the eutectic hydrated salts from 5.7 °C to 0.8 °C and increase the thermal conductivity from 0.82 W/(m·K) to 2.74 W/(m·K). They demonstrated that such EG-impregnated composites had excellent thermal insulation effect and held the potential to serve as advanced building materials. In another work, Fu et al. [94] utilized EG as the supporting material, employed glycine as the temperature regulator and prepared SAT composites through a melted mixing approach. With 12 wt% of EG, the hydrated salt composites showed no leakage after melting, a suitable phase change temperature of 47.14 °C, a phase change enthalpy higher than 200 J/g, a low supercooling degree of 1.49 °C and a high thermal conductivity of 6.4 W/(m·K). They demonstrated that such a composite held the potential for floor radiant heating.

Li et al. [93] highlighted the influence of EG on the phase change behavior of sodium sulfate decahydrate (SSD, Na_2_SO_4_·10H_2_O), which suffers from shortcomings such as supercooling, phase separation, low thermal conductivity and recalescence. They found that the addition of CMC, borax decahydrate and OP-10 could effectively suppress supercooling and phase separation (Figure 8c), but the DSC curves of the resultant composites showed drastic recalescence in the freezing process. After impregnating the mixture salts within the pores of EG, the recalescence disappeared as the EG provided an effective heat transfer path to conduct the released latent heat in a timely manner. Experimentally, they observed that the thermal conductivity of the EG composites reached 1.96 W/(m·K), which was 2.6 times higher than the mixed hydrated salts. Benefiting from the increased thermal conductivity compared with neat SSD, the heating and cooling time of the composites impregnated within EG were shortened by 36.2% and 44.5%, respectively.

### 4.5. Hydrated Salt–Biochar Composites

Biomass char, a unique carbon material derived from the organic matter of plants, animals, and microorganisms through carbonization, possesses low-cost advantages over other carbon materials. Raw materials for producing biochar are diverse and include agricultural waste such as straw, corn stalks, and husks, as well as fruit, forestry residues such wood chips, sawdust, livestock manure such as animal waste from cows, and organic components of municipal solid waste [95]. Biomass char retains the original morphology and structure of its raw materials, featuring numerous pores and abundance of oxygen-containing groups. After carbonization, it exhibits enhanced thermal conductivity and elevated solar absorptance. Abundant, low-cost biochar provides a facile, economical and environmentally friendly supporting material to enhance the thermophysical properties of hydrated salts.

By using bagasse as the raw material, Wang et al. [96] prepared biomass-derived carbon-encapsulated shape-stabilized sodium sulfate decahydrate (SSD) composites. As shown in Figure 9a, the biochar was obtained through pressing fresh sugarcanes into bagasse and drying the bagasse at 150 °C before final carbonization at 500 °C for 1 h under a purge of N_2_ gas. The bagasse biochar (BBC) possesses a low density, a large specific surface area and numerous pores with a diameter around 10 nm, and thus could be used as the carbon supporting material to adsorb melted SSD. With 9 wt% of biochar, the SSD–BBC composites demonstrated a latent heat storage capacity of 161.5 J/g, a thermal conductivity of 1.79 W/(m·K), a solar–thermal conversion efficiency higher than 80%, the disappearance of supercooling, and improved stability in terms of leakage and repeated heating/cooling cycles.

By employing the biochar converted from the biomass waste of watermelon rinds, Qian, Wang, and colleagues [74] reported the preparation of lamellar-structured SAT composites. As shown in Figure 9b, watermelon rind was subject to hydrothermal treatment at 220 °C, followed by hydraulic compression in a steel mold and freeze-drying to obtain compressed BDCs. The obtained biochar was impregnated within molten SAT/CMC under vacuum, and the resultant composite was further compressed within a mold before solidification. The melted hydrated salts were confined between the lamellar BDC sheets without leakage occurring, and the composites demonstrated a low supercooling degree of 0.9 °C, a latent heat storage capacity higher than 200 J/g, a solar–thermal conversion efficiency up to 80% and excellent thermal cycling stability.

### 4.6. Hydrated Salt–Activated Carbon Composites

Activated carbon, known as activated charcoal, is a form of porous carbon supporting material characterized by a high surface area to volume ratio. Activated carbon is typically produced by the pyrolysis of organic materials such as cheap waste biomass, followed by an activation process that increases the number of micro and mesopores on the carbon structure, enhancing its adsorptive properties. The activation can be done through purging high-temperature steam or carbon dioxide or using chemical agents such as phosphoric acid or potassium hydroxide [86]. Depending on the raw material and pyrolysis processes, activated carbon can be hydrophilic and its internal surface can be modified to tailor its interaction with hydrated salts. Compared to other minerals such as zeolites, alumina, metal organic frameworks and silica gels, activated carbon has higher thermal conductivity and cost advantages [97,98].

Bennici et al. [99] reported the use of bead activated carbon with a specific surface area of 1300 m^2^/g as the matrix for encapsulating magnesium sulfate hydrates (MgSO_4_·7H_2_O). The preparation process mainly involved slowly depositing the salt mixture inside the activated carbon through wet impregnation before filtering through a Buchner funnel, and finally, oven-drying. They demonstrated that the resultant composites could be used for high-performance thermochemical heat storage with a high energy storage density of 920 J/g. The activated carbon supporting material facilitates the filling of the composites within the reactor, improves the water adsorption and desorption performance, enhances heat transfer within the composites and homogenizes the temperature distribution of the composites. They demonstrated the feasibility of employing these thermochemical heat storage composites for house heating and producing hot water.

### 4.7. Hydrated Salt–Hybrid Carbon Composites

By modifying carbon materials with other functional additives, hybrid carbon materials can further enhance the original thermophysical properties of hydrated salts, provide extra means to tune the charging/discharging and phase change behaviors, and expand the application scope of PCMs [28,72,84,100]. For example, hybridizing carbon fillers with magnetic-response fillers or electrical conductors can allow for charging and storage of magnetic-thermal and electro-thermal energy within PCMs and enable the dynamic tailoring of the charging process through applying external magnetic or electrical fields.

Li et al. [64] utilized Fe_3_O_4_-modified CNTs to improve the thermal conductivity of solar salts encapsulated within SiO_2_ shells and provide the composites with solar–thermal and magnetic–thermal conversion capabilities (Figure 10a). The superparamagnetic Fe_3_O_4_ nanoparticles were in situ formed on the surfaces of MWCNTs through reducing Fe^2+^ and Fe^3+^ with ammonia. The Fe_3_O_4_-modified CNTs, together with SiO_2_ capsules, improved the thermal conductivity from 0.236 W/(m·K) for the neat solar salts to 0.483 W/(m·K) with 3 wt% loading. In contrast to neat solar salts that have poor solar absorptance and no response to magnetic fields, the composites can directly harvest solar-thermal energy and can be charged by applying alternating magnetic fields.

In addition to microencapsulation, Yu et al. [101] coated a transparent and superhydrophobic silicone gel onto the external surfaces of magnesium chloride hexahydrate (MgCl_2_·6H_2_O, MCH) composites to prevent the loss of crystallization water during repeated thermal energy storage processes. As shown in Figure 10b, the commercial melamine sponge was coated with GO through cyclic impregnation and the GO-decorated sponge enabled photothermal conversion, suppressed the supercooling and leakage of MCH and improved heat transfer during charging and discharging processes. In comparison with impregnation within 3D porous carbon, the GO-decorated sponge reduced the usage of carbon materials and significantly improved the mechanical flexibility of the PCM composites. In addition to solid–liquid phase change latent heat thermal storage, the PCM composites could stably undergo hydration–dehydration to store thermochemical energy owing to the anti-leakage encapsulation by the silicone gel layer.

In addition to solar–thermal and magnetic–thermal conversion, Li et al. [102] reported the fabrication of a multifunctional composite mesh charger that also enables the electro-thermal charging of PCMs. As shown in Figure 10c, such a multifunctional charger was prepared by coating Fe–Cr–Al meshes with a graphite layer and an external polydimethylsiloxane (PDMS) layer. The inner Fe–Cr–Al mesh is electrically conductive and can efficiently convert electrical energy into heat with low voltage input. The graphite coating layer can not only enable the absorption of broadband sunlight and its direct conversion into heat, but can also provide the desired rough surfaces to amply the surface wettability of the mesh. The composite mesh achieved a high solar absorptance of ~94%, a high electrical conductivity of 6622 S/cm, strong resistance to corrosion and outstanding high-temperature stability.

Different from conventional slow heat diffusion-based charging, the mesh charger, driven by gravity, can continuously track the melting interface, thus realizing fast charging within large-volume PCMs. This dynamic charging approach demonstrates rapid heat response and a stable rapid charging rate and can be charged by both low-voltage electricity and large-flux solar illumination. This method achieves a high solar-thermal phase change efficiency (~90.1%) and electro-thermal storage efficiency (~86.1%), while maintaining approximately 100% of the energy density of the PCM. They also demonstrated that the dynamic charging process could be further accelerated by applying external magnetic fields to directionally drive the movement of the mesh charger within the melted PCM [103,104]. Such dynamic charging is applicable to a wide range of inorganic and organic PCMs, offering a solution to overcome the dilemma of achieving fast charging rates and sacrificing the latent heat storage density encountered in thermal-enhanced PCM composites.

### 4.8. Comparison of Property Enhancements by Different Carbon Fillers

As summarized in Table 2, the incorporation of various carbon fillers at relatively low loadings can simultaneously increase thermal conductivity, reduce supercooling degree, mitigate leakage, and improve the cycling stability of different hydrated salts while larging retaining the high latent heat storage capacity of the original PCMs. In comparison, the enhancement effects of specific thermophysical properties vary for different carbon fillers.

In terms of enhancing the thermal conductivity of the composites, CNTs and EG have shown much stronger enhancement effect than other carbon fillers, which could be attributed to their intrinsic high thermal conductivities and their capabilities to form connected heat conduction paths within the PCM composites. Specifically, 1D carbon fillers such as CNTs and CFs have large aspect ratio features, which allow for the formation of percolated thermal conduction paths with low loadings of fillers. By contrast, the enhancement of thermal conductivity by GO, biochar, and activated carbon is not so obvious due to their intrinsically lower thermal conductivity and failures in forming connected heat-conduction networks within the bulk composites. Without being subject to high-temperature heating and crystallization processes, these carbon fillers do not possess highly ordered crystalline structures to efficiently conduct heat. Meanwhile, these carbon fillers most often have many defects that can strongly impede heat transfer. The stronger thermal conductivity enhancement from CNTs than CFs is related to the dispersion state of these fillers within the hydrated salts. Generally, carbon fillers with hydrophilic surfaces have shown good wetting with hydrated salts, homogeneous dispersion, and reduced interfacial thermal resistance. Although 1D or 2D carbon fillers can also form percolated networks with increasing loadings, strong interfacial thermal resistance among neighboring fillers is unavoidable. Three-dimensional carbon foams, such as EG sheets that have been subject to compression molding, can effectively reduce interfacial thermal resistance between carbon fillers due to its existing connected heat-conduction networks. 

In terms of improving leakage and cycling stability and reducing supercooling, it is critical to provide sufficient capillary confinement forces and ample porous space that can adsorb the melted hydrated salts. Hydrophilic foams or networks with abundant surface functional groups can provide strong capillary forces to trap melted salts and can form hydrogen bonds with water molecules inside the composites, thus preventing the loss of crystallization water during repeated heating/cooling cycles. Reducing the supercooling of hydrated salts by adding heterogeneous carbon fillers is mainly influenced by the contact angle between the PCM crystal and the nucleating agent. In general, the hydrophilic surfaces of carbon fillers can serve as the desired heterogenous nucleation sites to lower the supercooling degree. To this end, GO, biochar and activated carbons are more favored for reducing supercooling. Alternatively, introducing hydrophilic functional groups onto the surfaces of graphene and EG can suppress supercooling while significantly increasing the effective thermal conductivity. Therefore, a compatible interface is critical for the improvement of thermal conductivity, anti-leakage, cycling stability, and lowering supercooling.

In comparison with single-component carbon fillers, introducing other functional components has demonstrated the capability to further improve the thermophysical properties of PCM composites and add extra functionalities. Most importantly, hybrid functional fillers can offer unique means to dynamically tune the charging/discharging process and achieve unprecedent comprehensive performance improvements.

## 5. Thermal Energy Management Applications

### 5.1. Thermal Management of Electronic Devices

Normal function of electronic devices requires a stable suitable working temperature range. For example, at low temperatures, lithium ion batteries (LIBs) often suffer from reduced capacity and a sharply declined lifetime due to decreased Li^+^ diffusion within both the electrodes and electrolyte [105,106]. At high temperatures, LIBs also show a drop in capacity, increased internal resistance, degradation of the electrodes and electrolyte, and even safety risks such as thermal runaway (TR), which triggers a chain reaction that can result in catching fire or an explosion [107,108]. By undergoing reversible melting and solidification, hydrated salts can absorb excessive heat in a timely manner and release the latent heat when the temperature of the electronic device drops below the freezing point of the PCM, thus offering a facile method for the passive thermal management of electronic devices [10,29].

LIBs are recommended to be operated within the temperature range of −40–60 °C to achieve optimum performances. With mechanical abuse such as nail penetration or electrical abuse such as forming a short-circuit, TR may occur, suddenly releasing an enormous amount of heat, and inducing serious safety problems including smoke, fire and explosion. As shown in Figure 11a, Zhang’s group [109] employed SAT/EG composites for the two-stage thermal management of LIBs, namely a solid–liquid phase change at 58 °C and thermochemical heat absorption in the temperature range of 106–140 °C. After compounding SAT with 20 wt% of EG, the resultant composites showed a high thermal conductivity of 4.96 W/(m·K), and the interconnected EG networks could confine melted SAT inside. At the first stage, the melting of the SAT/EG composites with a heat storage density of 225.1 kJ/kg could effectively keep the temperature of LIBs below the safe operation limit. In comparison with thermal management with organic PCMs, it was found that the SAT/EG composites could achieve cooler and more uniform temperature distribution when cooling a 20-cell battery pack and had price and safety advantages [110]. At the second stage, the chemical decomposition of SAT/EG composites through losing crystallization water at elevated temperatures could absorb a large amount of heat (568.3 kJ/kg) and inhibit TR propagation. Theoretical modeling showed that the SAT/EG composites could completely prevent cascade TR when a single battery cell within a battery pack was penetrated with a nail. By contrast, if using commercial aerogel thermal insulation materials to wrap the battery, the cascade TR was delayed but not avoidable.

Photovoltaic (PV) panels are another type of popular electronic devices that require critical thermal management. The efficiency of commercial PV panels is typically lower than 25%, which means that massive amount of heat is generated during the solar–electrical conversion process. The generated heat significantly increases the panel temperature, leading to reduced electricity generation efficiency and the shortened lifetime of PV panels. Instead of adopting active cooling techniques such as water spray cooling or forced airflow circulation, as shown in Figure 11b, Wang’s team [111] reported passive cooling of commercial PV panels by attaching a hydrogel composite at the back. The hydrogel composite was prepared through in situ polymerization of polyacrylamide (PAM) in a dispersion of CNTs followed by freeze-drying and immersion in CaCl_2_ solution. The hygroscopic CaCl_2_ salts enabled the large-capacity adsorption of water vapor from the surrounding air at night. During the daytime, the hot PV panels heated up the hydrogel composites and induced the evaporation of the adsorbed water, which in turn brought down the temperature. CNTs were used to increase the emissivity of the cooling composites and enhance radiative cooling performances. Under one-sun illumination, the temperature of the PV panel was reduced by more than 10 °C. Outdoor field tests conducted in Saudi Arabia showed that after attaching the hydrogel composites, the electricity generation was increased by 19% and 13% in summer and winter, respectively. Through further integrating with an aluminum condensation chamber, they further demonstrated the capability to produce clean water from ambient air. The harvested water could be used for cleaning the dust on the PV panels or even for meeting personal drinking needs.

### 5.2. Thermal Management of Buildings

Building energy consumption accounts for about ~30% of the global energy budget and contributes ~30% of CO_2_ emissions. A large portion of thermal energy is used for heating, cooling and maintaining indoor temperatures in residential, commercial and public buildings. Incorporating low-cost hydrated salt PCMs into various parts of buildings, such as the roof, the wall and the floor, offers the opportunity to reduce energy consumption, enhance indoor thermal comfort and facilitate the usage of renewable thermal energy.

Wang et al. [96] conducted a proof-of-demonstration experiment by employing a composite material made of SSD hydrated salts and biomass charcoal on a roof to stabilize the temperature of buildings (Figure 12a). The PCM composite was enclosed in a square box made of acrylic board, surrounded by insulation materials that simulated the concrete used in practical buildings. To mimic daytime and night conditions, they turned on a xenon lamp on for 5000 s and turned off the lamp for 10,000 s. Compared with the pristine roof, the roof equipped with PCM composites had much smaller temperature fluctuation. They also carried out economic analyses on the static and dynamic payback period and pointed out that the inorganic hydrated salt/biochar composites were more economical than other PCM systems owing to the cost advantages from both hydrated salts and the biomass biochar.

In another work [112], researchers conducted economic analyses on employing hydrated salt composites for household heating (Figure 12b). They observed that it took a longer time to charge the PCM composites under low-intensity sunlight illumination conditions. The phase-change durations under 0.8 sun and 0.6 sun were 2610 s and 5540 s, respectively. Using hot water production as an example, they compared the costs of electricity consumption and the raw materials for fabricating the PCM composites and found that the solar-thermal PCM composites had a payback period of ~3 years.

In addition to roof heating and producing hot water, Zhang’s group developed a dual-layer radiant floor system using hydrated salt–EG composites for indoor thermal comfort regulation in winter and summer [113]. As shown in Figure 12c, the floor system had a heat storage layer made of Na_2_HPO_4_·12H_2_O–Na_2_SiO_3_·5H_2_O/EG composite (melting point: 31.3 °C), and a cold storage layer made of CaCl_2_·6H_2_O–NH_4_Cl-SrCl_2_·6H_2_O/EG composite (melting point: 20.2 °C). By placing the heat storage layer on top and the cold storage layer at bottom, the radiation floor doubled the thermal comfort duration of the test room when compared with the room only containing pebbles in the floor in the winter climate. In the summertime, the radiant floor with an upper cold storage layer and a lower heat storage layer provided a thermal comfort duration up to 8 h. Compared to the reference room that did not have the PCM radiant floor installed, it was estimated that the phase change room could save on electricity costs by more than 50% during winter and summer days.

### 5.3. Thermal Management of the Human Body

Due to the non-toxic and harmless nature of hydrate salts and their suitable temperature range, the use of composite materials for biological thermotherapy to alleviate local pain and promote blood circulation has become another important application direction. Hydrate salts have been compounded with other functional materials to prepare high-performance, flexible, wearable composites that can be charged with light, electricity or magnetic fields to heat up the human body, blood vessels or the air people breathe.

Xu and coworkers reported the preparation of photothermal hydrogel composites by using SAT as the PCM, GO as the photothermal agent, and acrylamide and konjac glucomannan as the supporting structure [112]. The hydrogel networks simultaneously reduce the leakage of melted SAT, suppress supercooling, enhance heat transfer and afford the composites with mechanical flexibility. The incorporation of GO enables the charging of the PCM composites with direct solar illumination, with a high conversion efficiency of 89.7%. In comparison to traditional heating techniques using hot water or hot towels, PCM composites can better retain the thermotherapy temperature within a narrow range and prolong the heat-releasing duration. As shown in Figure 13a, the SAT–hydrogel composite that was charged by photothermal conversion to 55 °C could be directly attached onto human skin and provide stable thermotherapy for 5 min.

To pursue personal thermal management applications and overcome the drawbacks of pristine hydrated salts, Liu et al. [114] prepared self-healing flexible hydrogel composites by blending SAT with acrylamide, aqueous starch and graphite followed by subsequent radical initiated polymerization of the acrylamide. They pointed out that the addition of graphite not only effectively addressed supercooling issues, but also enhanced the cyclic performance of the composite. The flexible hydrogel networks allowed the composite to undergo 40% deformation and the rich hydrogen bonding within the composites enabled the self-healing capability. The flexibility and self-healing of the hydrogel–SAT composites facilitated their integration with wearable textiles for thermal management of the human body. As shown by the infrared images in Figure 13b, the composites could be attached onto the back of the human hand or the arm, providing thermal comfort at temperatures above 40 °C for more than 15 min owing to the high latent heat storage capacity.

## 6. Summary and Outlook

This work overviews the recent progress in designing and fabricating carbon-enhanced hydrated salt PCM composites that overcome the shortcomings of pristine hydrated salts, such as low thermal conductivity, supercooling, phase separation, poor cycling stability and leakage, and enable the direct conversion of other forms of energy into storable heat, thereby enabling their thermal management applications.

Although substantial improvement in their thermophysical properties had been achieved with the addition of various carbon fillers, how to rationally design the fillers to optimize the comprehensive performance of the hydrated salt PCM composite is still a grand challenge. It requires continuous research efforts to gain a deeper understanding of the enhancing mechanisms, exploring advanced fabrication and characterization tools, and expanding the application fields of hydrated salt PCM composites. It is recommended that the research community working on carbon-enhanced inorganic PCM composites can take full advantage of the fast development of organic PCM composites. For example, there are several comprehensive review articles on the design, fabrication, performance and applications of carbon-enhanced organic PCM composites [115,116,117], which can provide valuable guidelines and inspiration to accelerate the development of high-performance hydrated salt composites. In particular, the classical theoretical models that predict the dependence of the effective thermal conductivity of organic PCM composites on the intrinsic thermal conductivity of carbon fillers; the shape, size, loading and distribution of carbon fillers; and the formation of connected networks can shed light on the design and construction of inorganic hydrated salt PCM composites [118,119].

So far, the reported thermal conductivity values of the hydrated salt composites are much lower than the expected values. Limited interfacial heat transfer might be the dominant factor in this difference due to the inevitable interface resistance between the carbon fillers, and the carbon fillers and the PCM matrix. Most often, diverse thermophysical properties or functionalities have different and even contradictory requirements for the added carbon fillers. For instance, increasing the loading of the carbon fillers can help improve the thermal conductivity of the composites, but it sacrifices the specific latent heat storage capacity. The microstructure and surface chemistry of carbon fillers are two powerful tools we can rely on to design hydrated salt PCM composites and systematically tailor their thermophysical properties. Current findings on the influence of filler parameters, including the morphology, size, shape, loading, surface functional groups and the intrinsic properties of each carbon filler, can be used as inputs to establish a comprehensive model to provide full-picture analyses of the thermophysical properties/performance of carbon-enhanced hydrated salt composites. To this end, advanced simulation tools, such as machine learning techniques, might be able to help accelerate the identification process.

Regarding fabrication methods, most of the investigated approaches are for laboratory-scale trials. Developing suitable preparation techniques and controlling the quality of large-size composites is urgently needed for their subsequent scaled-up manufacturing. After identifying the ideal chemical composition, systematic investigation into the key parameters of the fabrication process, such as the blending rate, the compression pressure, and the impregnation vacuum level, on the homogeneity and repeatability of the PCM composites should be carried out.

To date, the characterization of thermophysical properties has been focused on static macroscale measurements that have been carried out in the short term. Practical applications, such as domestic heating or the thermal management of electronic devices, require evaluation of the performance of PCM composites over long periods and under extreme operation conditions. To meet the application needs, on one hand, more detailed characterization of the heat transfer process and the solid–liquid phase transition behavior at fine scales with tools such as high-speed infrared cameras can provide deeper insights into the governing enhancement mechanisms. On the other hand, carrying out continuous dynamic measurements of the key thermophysical properties of hydrated salt composites over long durations under simulated operation conditions, for example with controlled temperature fluctuations and different humidity levels, can pave the way for the practical application of fabricated composites.

Corrosion has been viewed as another key issue that limits long-term reliability and the practical applications of hydrated salts. Although the corrosion behavior of various hydrated salts in different types of metallic containers have been investigated [120,121], the chemical compatibility between carbon-enhanced hydrated salt composites and containers has rarely been reported. In comparison with pristine hydrated salts, one obvious advantage of the carbon-enhanced composites is that they are leakage-proof, which significantly reduces the probability for direct contact between the corrosive salts and the containers. Future research directions can be focused on further improving the tightness of the PCM composites or enhancing the corrosion resistance of the container through applying specialized coatings on its surface.

Currently investigated applications are mainly related to the harvesting of renewable thermal energy and temperature control of buildings, electronic devices and human bodies. After overcoming the shortcomings with carbon fillers and achieving long-term stability, it is anticipated that hydrated salt composites should have the potential for a broader range of applications beyond what have been explored so far. More interdisciplinary research and development is encouraged to be carried out to promote widespread applications. For example, considering the biocompatibility of hydrated salts and their suitable phase change temperatures, carbon-enhanced hydrated salt PCM composites might find use in popular biomedical applications, such as therapeutic hypothermia, surgical procedures, regulated drug delivery, wound dressings, cold chain logistics, medical imaging, hyperthermia treatment, prosthetics and orthotics, space and military medicine and cardiac and vascular procedures [122,123,124].

## Figures and Tables

**Figure 1 nanomaterials-14-01077-f001:**
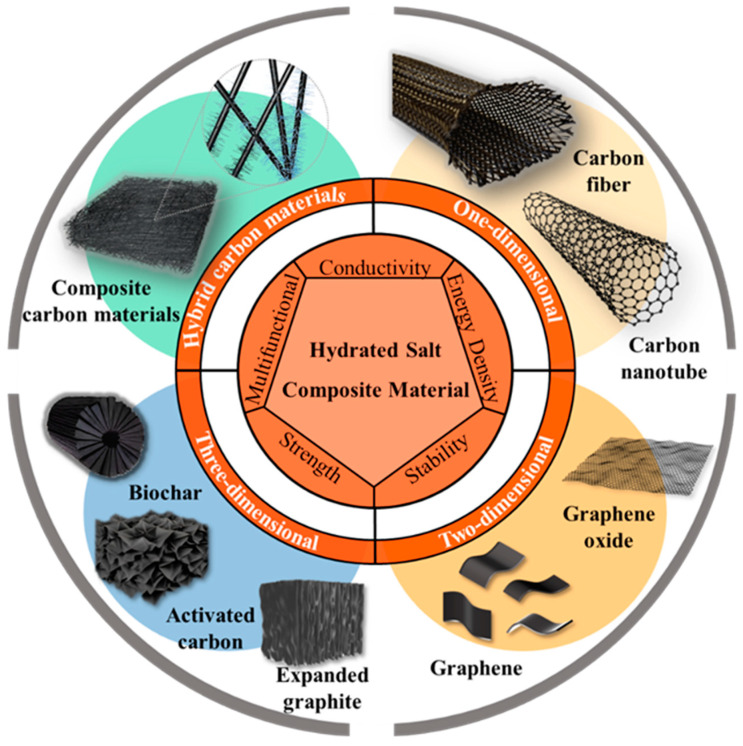
Hydrated salt PCMs enhanced by various functional carbon materials.

**Figure 2 nanomaterials-14-01077-f002:**
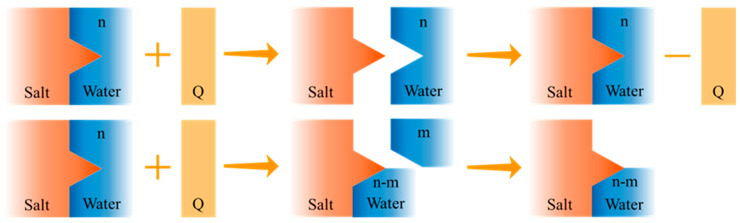
Schematic showing the congruent and incongruent melting processes of hydrated salts.

**Figure 3 nanomaterials-14-01077-f003:**
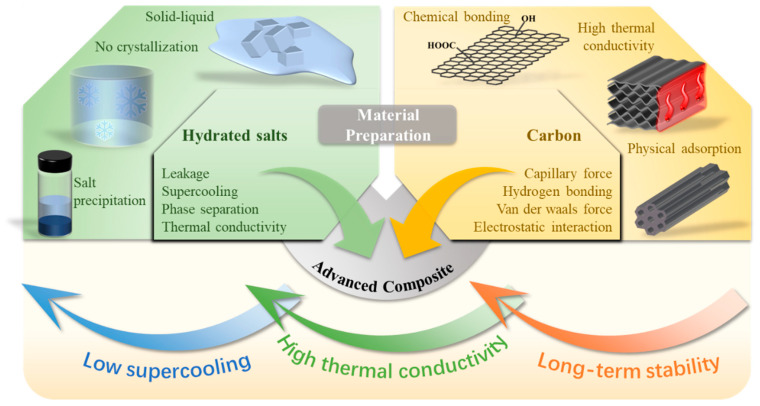
Preparation of carbon-enhanced PCM composites to overcome the shortcomings of hydrated salts.

**Figure 4 nanomaterials-14-01077-f004:**
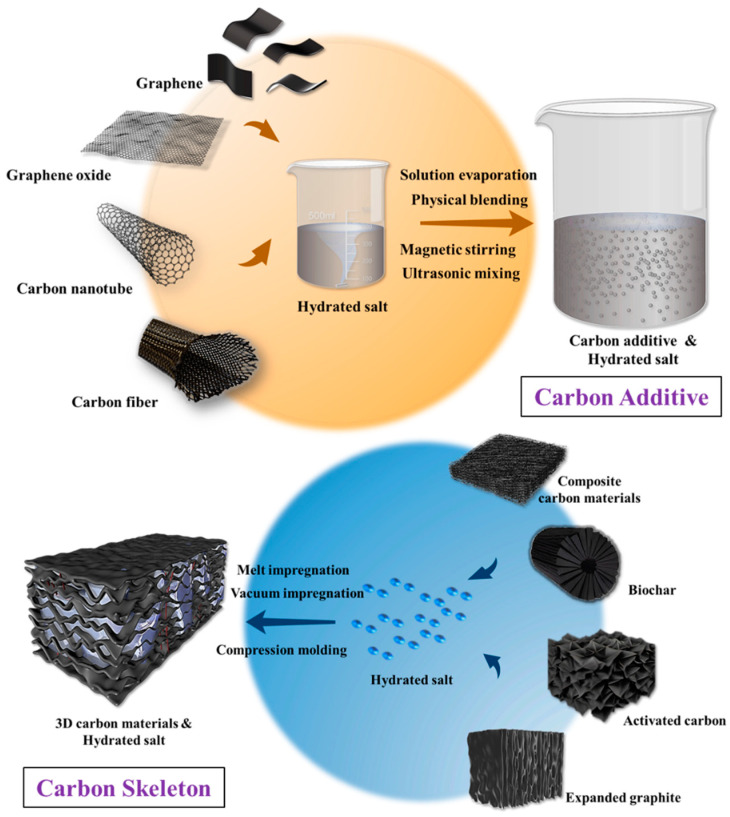
Routes for preparing carbon-enhanced hydrated salt PCM composites: carbon additive method and carbon skeleton method.

**Figure 5 nanomaterials-14-01077-f005:**
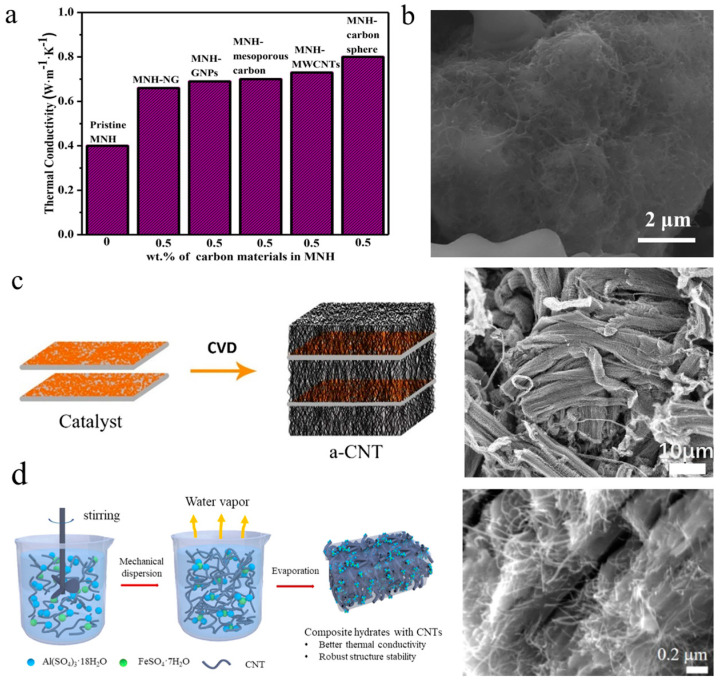
(**a**) Comparison of thermal conductivity enhancement of magnesium nitrate hexahydrate by carbon spheres, MWCNTs, mesoporous carbon, graphene nanoplatelets and nano graphite with the same loading of 0.5 wt% (Figure 5a reprinted/adapted with permission from Ref. [78]. 2021, Elsevier Ltd.) (**b**) SEM image showing uniform dispersion of hydrophilic CNTs within eutectic salts of sodium acetate trihydrate/sodium monohydrogen phosphate dodecahydrate (Figure 5b reprinted/adapted with permission from Ref. [79]. 2021, Elsevier Ltd.) (**c**) Schematic route for synthesizing aligned CNTs (a-CNTs) and a representative SEM image of a-CNT (Figure 5c reprinted/adapted with permission from Ref. [80]. 2019, Elsevier Ltd.) (**d**) Binary hydrated salts of Al_2_(SO_4_)_3_·18H_2_O/FeSO_4_·7H_2_O with a mass ratio of 2:1 enhanced by a-CNTs (Figure 5d reprinted/adapted with permission from Ref. [81]. 2021, the American Chemical Society).

**Figure 6 nanomaterials-14-01077-f006:**
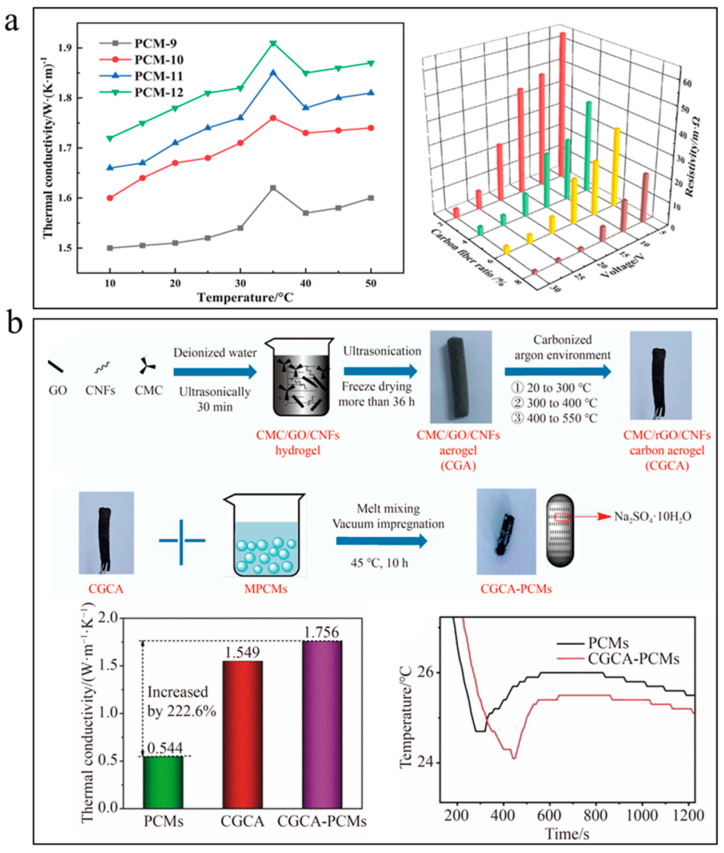
(**a**) Improving thermal conductivity and decreasing electrical resistance of Na_2_SiO_3_·9H_2_O composites with short-cut CFs (Figure 6a reprinted/adapted with permission from Ref. [84]. 2022, Wiley-VCH GmbH). (**b**) GO–CF aerogel loaded with eutectic salts of Na_2_SO_4_·10H_2_O and Na_2_CO_3_·10H_2_O to increase thermal conductivity and reduce supercooling (Figure 6b reprinted/adapted with permission from Ref. [85]. 2022, Springer Nature).

**Figure 7 nanomaterials-14-01077-f007:**
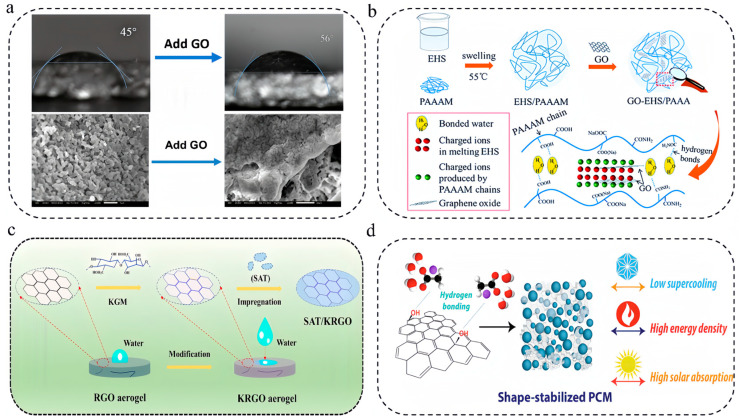
(**a**) Addition of GO enables a reduction in the contact angle between melted DHPD and the porous expanded vermiculite skeleton from 56° to 45° (Figure 7a reprinted/adapted with permission from Ref. [67]. 2020, the American Chemical Society). (**b**) Improving thermophysical properties of eutectic hydrated salts by a GO-modified poly(acrylamide-co-acrylic acid) copolymer (PAAAM) hydrogel (Figure 7b reprinted/adapted with permission from Ref. [69]. 2016, Royal Society of Chemistry). (**c**) Impregnation of SAT within super-hydrophilic rGO aerogels modified by konjac glucomannan (Figure 7c reprinted/adapted with permission from Ref. [87]. 2022, the American Chemical Society). (**d**) Shape-stabilized solar-thermal storage SAT composites enhanced by hydrophilic GNPs and sodium phosphate monobasic monohydrate nucleating agents (Figure 7d reprinted/adapted with permission from Ref. [88]. 2021, Elsevier).

**Figure 8 nanomaterials-14-01077-f008:**
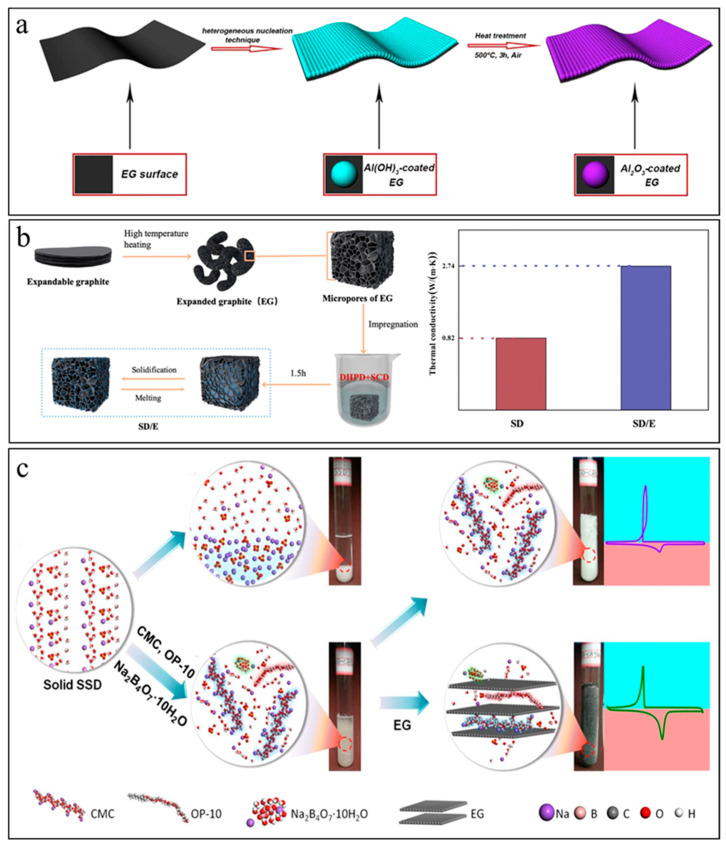
(**a**) Coating EG with an Al_2_O_3_ layer to achieve surface hydrophilicity and improve wetting with hydrated salts (Figure 8a reprinted/adapted with permission from Ref. [91]. 2018, Elsevier). (**b**) Reducing supercooling and increasing thermal conductivity of eutectic hydrated salts by EG networks (Figure 8b reprinted/adapted with permission from Ref. [92]. 2021). (**c**) EG networks provide hydrated salts with an effective heat transfer path to conduct released latent heat in a timely manner and suppress recalescence during solidification (Figure 8c reprinted/adapted with permission from Ref. [93]. 2020, Elsevier).

**Figure 9 nanomaterials-14-01077-f009:**
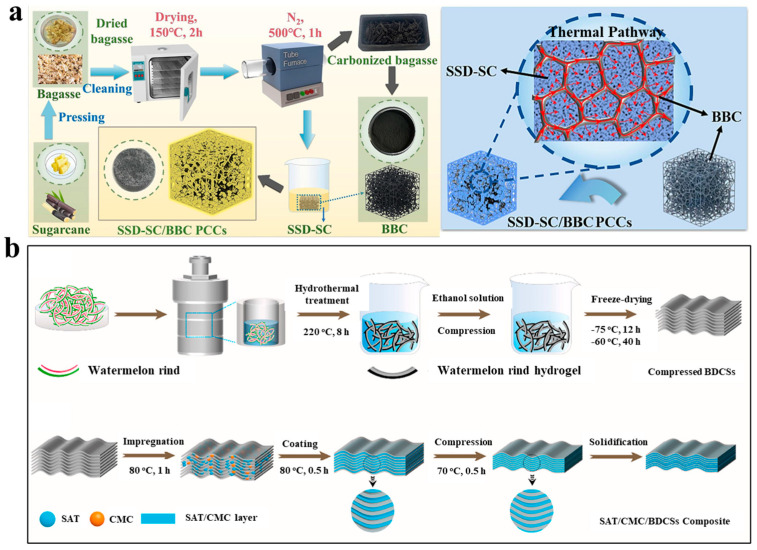
(**a**) Biomass-derived carbon-encapsulated shape-stabilized sodium sulfate decahydrate (SSD) composites (Figure 9a reprinted/adapted with permission from Ref. [96]. 2024, Elsevier). (**b**) Preparation of compressed biochar from watermelon rind and a biochar–hydrated salt composite. (Figure 9b reprinted/adapted with permission from Ref. [74]. 2020, Elsevier).

**Figure 10 nanomaterials-14-01077-f010:**
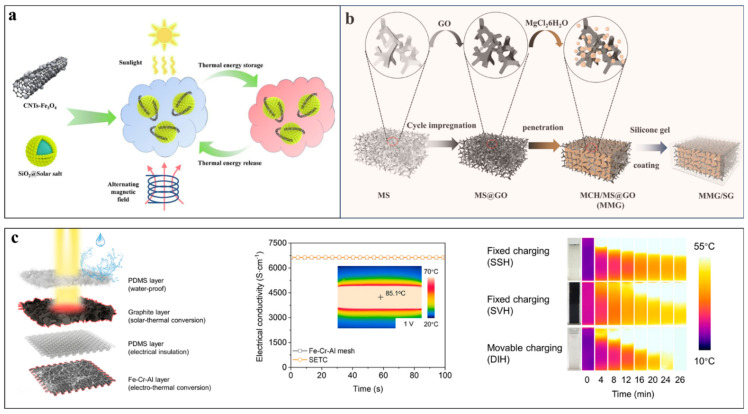
(**a**) Compounding SiO_2_ solar salts with Fe_3_O_4_-modified CNTs for simultaneous harvesting of solar-thermal and magnetic-thermal energy (Figure 10a reprinted/adapted with permission from Ref. [64]. 2023, Elsevier). (**b**) Magnesium chloride hexahydrate (MgCl_2_·6H_2_O, MCH) impregnated within a GO-decorated sponge for latent heat and thermochemical storage (Figure 10b reprinted/adapted with permission from Ref. [101]. 2023, Springer nature). (**c**) Dual-functional Fe–Cr–Al mesh coated with nano-graphite for movable charging of solar-thermal energy and electro-thermal energy (Figure 10c reprinted/adapted with permission from Ref. [102]. 2023, Elsevier).

**Figure 11 nanomaterials-14-01077-f011:**
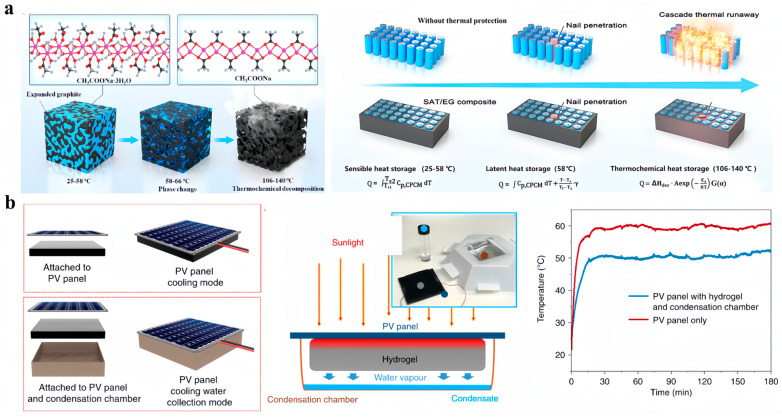
(**a**) SAT/EG composites for the two-stage thermal management of LIBs (Figure 11a reprinted/adapted with permission from Ref. [109]. 2022, Elsevier). (**b**) Passive cooling of commercial PV panels by attaching a polyacrylamide hydrogel loaded with CNTs and CaCl_2_ to the back side (Figure 11b reprinted/adapted with permission from Ref. [111]. 2020, Springer nature).

**Figure 12 nanomaterials-14-01077-f012:**
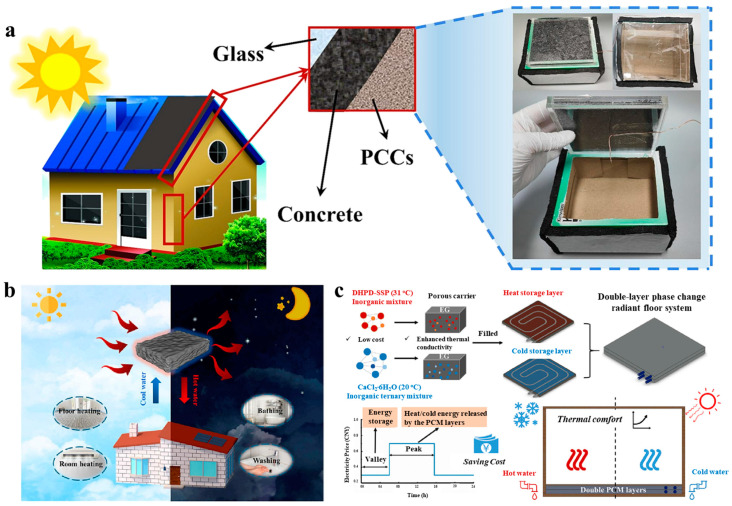
(**a**) SSD–biochar hydrated salt composites incorporated within a roof for stabilizing building temperature (Figure 12a reprinted/adapted with permission from Ref. [96]. 2024, Elsevier). (**b**) Hydrated salt composites for household heating (Figure 12b reprinted/adapted with permission from Ref. [112]. 2023, Elsevier). (**c**) A radiant floor system consisting of a heat storage layer made of Na_2_HPO_4_·12H_2_O–Na_2_SiO_3_·5H_2_O/EG composite and a cold storage layer made of CaCl_2_·6H_2_O–NH_4_Cl-SrCl_2_·6H_2_O/EG composite for improving indoor thermal comfort (Figure 12c reprinted/adapted with permission from Ref. [113]. 2020, Elsevier).

**Figure 13 nanomaterials-14-01077-f013:**
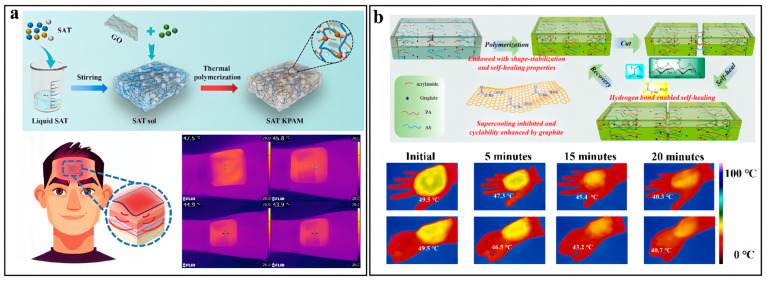
(**a**) An SAT–hydrogel composite attached onto human skin to provide stable thermotherapy for 5 min. (Figure 13a reprinted/adapted with permission from Ref. [112]. 2023, Elsevier). (**b**) Preparation of self-healable SAT–hydrogel composites that could be attached onto the back of a human hand or arm, providing thermal comfort for 20 min (Figure 13b reprinted/adapted with permission from Ref. [114]. 2023, Elsevier).

**Table 1 nanomaterials-14-01077-t001:** Thermophysical properties of representative hydrated salt phase change materials.

Hydrated Salt	Melting Temperature (°C)	Heat of Fusion (J/g)	Thermal Conductivity (W/m·K)	Specific Heat (Solid) (J/kg·K)	Density (Solid) (g/cm^3^)	Ref.
LiClO_3_·3H_2_O	8	253			1.72	[39,40]
KF·4H_2_O	19	231		1.84	1.45	[41]
Mn(NO_3_)_2_·6H_2_O	25.8	125.9			1.6	[42]
CaCl_2_·6H_2_O	28	174	1.088	1.42	1.8	[1,39,40]
LiNO_3_·3H_2_O	30	256				[43]
Na_2_SO_4_·10H_2_O	32.4	248	0.5		1.49	[44,45]
Na_2_CO_3_·10H_2_O	33	247	0.45	1.88		[42,46]
Na_2_HPO_4_·12H_2_O	35–44	280	0.47	1.7		[39,42,47]
Zn(NO_3_)_2_·6H_2_O	36	149.6		1.34	1.94	[42]
FeCl_3_·6H_2_O	37	223				[48]
CoSO_4_·7H_2_O	40.7	170				[40,49]
Fe(NO_3_)_2_·9H_2_O	47	155				[40]
Na_2_SiO_3_·4H_2_O	48	168				[40]
K_2_HPO_4_·7H_2_O	48	99				[40]
MgSO_4_·7H_2_O	48.5	202				[42]
Na_2_S_2_O_3_·5H_2_O	49	220	0.5		1.75	[46,50]
Ni(NO_3_)_2_·6H_2_O	57	169				[42]
CH_3_COONa·3H_2_O	58	226–264	1.97		1.45	[51]
MgCl_2_·4H_2_O	58	178				[40]
NaOH·H_2_O	64.3	273				[40]
Na_3_PO_4_·12H_2_O	65–69	190				[40]
Ba(OH)_2_·8H_2_O	78	266	0.65			[52]
Na_4_P_2_O_7_·10H_2_O	79.5	184	1.82			[53]
Mg(NO_3_)·6H_2_O	89	162	1.81		1.636	[42]
Al(NO_3_)_2_·8H_2_O	89.3	150				[42]
Mg(NO_3_)·6H_2_O	89.9	163	1.81	0.669	1.636	[49]
KAl(SO_4_)_2_·12H_2_O	91	184				[49]
NH_4_Al(SO_4_)_2_·12H_2_O	95	269	1.71		1.65	[42]
MgCl_2_·6H_2_O	117	169	0.69		1.56	[40]

**Table 2 nanomaterials-14-01077-t002:** Enhancement of the thermophysical properties of hydrated salts by various carbon fillers.

Hydrated Salt	Filler	PCM (wt%)	Carbon (wt%)	Theral Conductivity (W/m·K)	Supercooling Degree (°C)	Leakage	Tm (°C)	ΔH (J/g)	Ref.
Mg(NO_3_)_2_·6H_2_O	CNT	99.5	0.5	0.73		Not mentioned	90.8	141.4	[78]
Na_2_HPO_4_·12H_2_O	CNT	94.55	1.25			Not mentioned	44.32	216.32	[79]
Al_2_(SO_4_)_3_·18H_2_O/ FeSO_4_·7H_2_O (2:1)	CNT	94	5	3.23		No leakage	98.74	422.4	[81]
CH_3_COONa·3H_2_O	CNT	90	2	7.178	0.6	No leakage	55	177.6	[82]
Na_2_HPO_4_·12H_2_O	CF	72	8	1.91		No leakage			[84]
Na_2_SO_4_·10H_2_O/ Na_2_CO_3_·10H_2_O (9:1)	CF	96.9	3.1	1.756	0.7	No leakage	31.4	157.1	[85]
Na_2_HPO_4_·12H_2_O	GO	95.4	0.3			Not mentioned	42.5, 48.8	229	[67]
Na_2_CO_3_·10H_2_O/ Na_2_HPO_4_·12H_2_O (2:3)	GO	92	3	1.1	8	No leakage	23	200.3	[69]
CH_3_COONa·3H_2_O	GO	98.5	0.15		1.9	No leakage	57.83	252.8	[87]
MgCl_2_·6H_2_O- NH_4_Al(SO_4_)_2_·12H_2_O	EG	81.9	15	4.789		No leakage	23.5	196.2	[91]
Na_2_CO_3_·10H_2_O	EG	85.5	5	2.74	0.8	No leakage	31.52	123.6	[92]
CH_3_COONa·3H_2_O	EG	88	12	6.4	1.49	No leakage	47.14	200	[94]
Na_2_SO_4_·10H_2_O	Biochar	83	9	1.79		No leakage	30.2	161.5	[96]
CH_3_COONa·3H_2_O	Biochar			1.17	0.9	No leakage	61.1	209.8	[74]

## Data Availability

The original contributions presented in the study are included in the article, further inquiries can be directed to the corresponding author.

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
