# Peer review of "Carbon-Enhanced Hydrated Salt Phase Change Materials for Thermal Management Applications"

_nanomaterials, 2024, doi:10.3390/nano14131077_

Round 1

Reviewer 1 Report

Comments and Suggestions for Authors

The manuscript provides a comprehensive overview of the improvements in using hydrated salts as phase change materials by integrating carbon materials. These carbon additions help overcome inherent limitations such as low thermal conductivity and issues related to phase stability and leakage, enhancing the practical applications of these composites in managing thermal energy efficiently. Minor revision is suggested. There are a few suggestions that the authors may consider:

1. The first three paragraphs of the introduction provide a broad overview, which could be more impactful if tightly focused on carbon-enhanced phase change materials (PCMs). It would be beneficial to directly address the specific advantages of these PCMs early on to set the stage for the detailed discussions that follow. Clarifying the unique benefits of carbon-enhanced PCMs would help readers understand their significance and applications more clearly.

2. The manuscript would greatly benefit from an expanded discussion on the underlying physics and fundamental mechanisms through which different carbon materials enhance the thermophysical properties of hydrated salts. A detailed exploration of recent theories and proposed mechanisms would enrich the readers’ understanding and highlight the innovative aspects of current research in this field. This addition would provide a more comprehensive scientific context to the advancements discussed.

Comments on the Quality of English Language

The quality of English is generally good.

Author Response

  1. Reply: We thank the reviewer for the kind suggestion. In the revised manuscript, we merged the first two paragraphs to shorten the background introduction on thermal energy application needs. Following the reviewer’s suggestion, we then directly point out the advantages and the shortcomings of pristine hydrated salts, and briefly highlight how various carbon materials can help improve thermophysical properties and the unique benefits from the carbon-enhanced PCMs.
  2. Reply: We thank the reviewer for the kind suggestion. Following the reviewer’s suggestion, we added another section (4.6 Comparison of property enhancement by different carbon fillers) to comprehensively compare the enhancement of key thermophysical properties and discuss the corresponding mechanisms for different carbon fillers.

Reviewer 2 Report

Comments and Suggestions for Authors

The article is an interesting review on a very timely and valuable topic. The authors carefully review the research available in the literature and draw conclusions that may have important practical implications. However, it would be worthwhile to go further: 

1. The purpose of the review is not sufficiently emphasised. The same applies to the conclusions, which could be strengthened.

2. The references section is not properly edited.

Comments on the Quality of English Language

Moderate editing of English language required.

Author Response

  1. Reply: Thanks for the kind suggestion. In the revised manuscript, we shorten the description on general background and added more contents on carbon-enhanced hydrated salt PCM composites. In this revised version, we pointed out the research gap in this research field and clarified the purpose for this review. In the conclusion section, we expanded the discussion on future research needs and potential directions.
  2. Reply: Thanks for the comment. We thoroughly double-check all the references and made appropriate editorial changes in the revised manuscript.

Reviewer 3 Report

Comments and Suggestions for Authors

In this work, authors have provided a comprehensive review over the development of carbon-enhanced hydrated salt PCM composites. They firstly introduced the solid-liquid phase change process, the advantageous features, and the shortcomings of hydrated salts for storage and management of thermal energy. Then briefly described how various carbon materials can enhance the thermophysical properties of hydrated salts and the typical methods for fabricating carbon-enhanced hydrated salt PCM composites. By examining representative examples, authors discussed how the intrinsic properties of various carbon materials such as thermal conductivity, surface chemistry, morphology, microstructure, and their loading, dispersion affect the performances of the resultant hydrated salt composites including heat transfer behavior, reversible solid-liquid phase change stability, anti-leakage, solar-thermal conversion, and additionally-afforded functionalities. Please find comments as below;

1. Title does not sound clear to represent the insightful purpose of this review article which should be amended based on objectives of the study. 

2. Introduction section should elucidate the existent research gap and the necessity of this review article to contribute to the subject area and which areas of knowledge have not covered by literature which are going to be addressed through this study. Each reference should be discussed separately with main focus on critical analysis. 

3. More salts could be included in table 1 with consideration of more therm0physical properties such cp.

4. Comprehensive comparative study is required regarding the phase transition process of organic PCMs and hydrated salts to discuss differences in details.

5. Chemical reaction among carbon based hydrated salts in terms of stability aspects should be discussed as this parameter mostly impacts the ultimate thermophysical properties. 

6. Figure 4 should be elaborated as there are more available approaches. 

7. Figure 5; Supporting the comparison with a comprehensive table would be better way to comprehend. 

8. Figure 6; Scientific reasons behind these phenomenon improvements should be discussed as well. 

9. A comparative study among different types of carbon based additives towards hydrated salt should be elaborated in clearer manner. 

10. Challenges should be taken into account precisely.

11. Comprehensive proofread is essential throughout the manuscript. 

Comments on the Quality of English Language

Typo/grammatical errors should be rectified throughout the manuscript. 

Author Response

  1. Reply: Thanks for the kind suggestion. In the revised manuscript, we revised the title and highlight the purpose in the last paragraph of the introduction section.
  2. Reply: Thanks for the kind suggestion. In the revised manuscript, we pointed out the research gap in this research field and clarified the purpose for this review.

  3. Reply: Thanks for the kind suggestion. In the revised manuscript, we added more salts along with their specific heat in Table 1.

  4. Reply: Thanks for the kind suggestion. In the revised manuscript, we added descriptions on the solid-liquid phase transition to highlight the difference and the similarity as well.

  5. Reply: Thanks for the kind suggestion. In the revised manuscript, we added brief discussion in the carbon-enhancement strategy section to introduce how carbon fillers can improve both the shape stability and cycling stability of hydrated salts.

  6. Reply: Thanks for the comments. Herein, we divide the general route into two categories: the additive route and the skeleton route, which is based on the distribution state of carbon fillers within hydrated salts. We agree that there are more available fabrication methods for preparing such carbon-enhanced PCM composites. In the revised manuscript, we add more preparation methods such as solution evaporation, melted blending, melted impregnation.

  7. Reply: Thanks for the kind suggestion. In the revised manuscript, we added Table 2 to comprehensive summarize the enhancement of thermophysical properties by various carbon fillers and provided corresponding comparison on the enhancement effect.

  8. Reply: Following the suggestion, we added the corresponding reasons that lead to the improvement of thermal conductivity, supercooling and anti-leakage properties.

  9. Reply: Thanks for the comment and suggestion. In the revised manuscript, we added another section (4.6 Comparison of property enhancement by different carbon fillers) to clearly elaborate the enhancement effect.

  10. Reply: Thanks for the comment and suggestion. In the revised manuscript, we added more description on the future challenges and potential research direction in the summary and outlook section.

  11. Reply: Thanks for the suggestion. We have carefully edited the whole manuscript.

Reviewer 4 Report

Comments and Suggestions for Authors

Recommendation: Minor revision

Comments for the revision:

1.      In my opinion, it is necessary to reform abstract and make it more understandable and transparent. It is necessary to present the obtained results in the abstract. Please use numeric values. Please refer to the results of these studies in the summary.

2.      Moreover, remove the word “we” in the abstract. What was the major outcomes from the review work?

3.      Since the presented material is to broaden the state of knowledge, please define what conclusions the presented research, compiled in one literature review, leads to. A critical comment based on the presented research results and numerical values is necessary.

4.      Please clearly declare how the presented article extends the current state of knowledge as compared to other review works in the same context.

5.      The summary of this paper is very comprehensive and detailed, and some of the viewpoints are still very meaningful. It is recommended to further add more detailed analysis, such as the study of phase change microcapsule suspensions at micro scales in medical applications.

6.      It is recommended to add a section titled “Limitations and research scope for the future”.

Comments on the Quality of English Language

Minor editing is needed

Author Response

  1. Reply: Following the reviewer’s suggestion, we revised the abstract accordingly.
  2. Reply: Following the reviewer’s suggestion, “we” has been removed from the revised abstract. The major outcomes from this manuscript are the summary on rational design, fabrication, enhancement mechanisms of hydrated salt PCMs by various types of carbon fillers and the applications of the obtained composites for thermal management. We clarify these in both the abstract and conclusion sections.

  3. Reply: Following the reviewer’s suggestion, we add a separate section (4.6 Comparison of property enhancement by different carbon fillers) to comprehensively comment on the present research results.

  4. Reply: Following the reviewer’s suggestion, we add description about the research gaps in the introduction section and the future research needs in the conclusion section. These added descriptions can help clarify how this work can help the research community working on carbon-enhanced hydrated salt PCM composites.

  5. Reply: Following the reviewer’s suggestion, we expanded the summary section by adding more discussion on current challenges and future research directions. In terms of applications, we highlight the needs and bright future to explore biomedical applications.

  6. Reply: Thanks for the suggestion. In the revised manuscript, we added more discussion the limitations of current research progress, and pointed out possible directions to further push the improvement of thermophysical properties and practical applications.

Round 2

Reviewer 3 Report

Comments and Suggestions for Authors

Authors have addressed all raised concerns properly and the current version of the manuscript is acceptable. 

Author Response

Comment 1: Authors have addressed all raised concerns properly and the current version of the manuscript is acceptable. 

Response 1: We sincerely thank the reviewer for all the helpful comments and suggestions.